# Effects of Different Calcium Silicate Cements on the Inflammatory Response and Odontogenic Differentiation of Lipopolysaccharide-Stimulated Human Dental Pulp Stem Cells

**DOI:** 10.3390/ma12081259

**Published:** 2019-04-17

**Authors:** Minsun Chung, Sukjoon Lee, Dongzi Chen, Ukseong Kim, Yaelim Kim, Sunil Kim, Euiseong Kim

**Affiliations:** 1Microscope Center, Department of Conservative Dentistry and Oral Science Research Center, Yonsei University College of Dentistry, 50-1 Yonsei-Ro, Seodaemun-Gu, Seoul 03722, Korea; mschung@yuhs.ac (M.C.); dongzi-chen@yuhs.ac (D.C.); ukpower@yuhs.ac (U.K.); ylk001@yuhs.ac (Y.K.); 2BK21 PLUS Project, Yonsei University College of Dentistry, 50-1 Yonsei-Ro, Seodaemun-Gu, Seoul 03722, Korea; shatoa@yuhs.ac; 3Department of Electrical & Electronic Engineering, Yonsei University College of Engineering, 50 Yonsei-Ro, Seodaemun-Gu, Seoul 03722, Korea

**Keywords:** calcium silicate cement, human dental pulp stem cell, lipopolysaccharide, inflammation, odontogenic differentiation

## Abstract

This study aimed to analyze the effects of different calcium silicate cements (CSCs) on the inflammatory response and odontogenic differentiation of lipopolysaccharide-stimulated human dental pulp stem cells. Human dental pulp stem cells (hDPSCs) were stimulated with lipopolysaccharide (LPS) to induce inflammation. These LPS-induced dental pulp stem cells (LDPSCs) were cultured with ProRoot MTA, Biodentine, Retro MTA, and Dycal. Cell viability was evaluated using the Cell Counting Kit-8 assay. Interleukin (IL)-6, IL-8, and transforming growth factor (TGF)-β1 cytokine levels were assessed using the enzyme-linked immunosorbent assay. The expressions of alkaline phosphatase (ALP), osteocalcin, and runt-related transcription factor 2 (RUNX2) were analyzed through real-time polymerase chain reaction. ProRoot MTA, Biodentine, and Retro MTA did not significantly decrease the cell viability of LDPSCs for up to 48 h (*p* < 0.05). Retro MTA significantly decreased the expression of IL-6 and IL-8 by LDPSCs. ProRoot MTA and Biodentine significantly reduced TGF-β expression by LDPSCs (*p* < 0.05). Regarding odontogenic differentiation, all CSCs had no effect on ALP expression but increased the production of RUNX2 at 12 h.

## 1. Introduction

Calcium silicate cements (CSCs) are used in various types of treatments in the endodontic field. These treatments include pulp capping, regenerative therapy, apexification, and apical surgery. CSCs are known to have superior biocompatibility, sealing ability, and hard tissue induction potential compared to previous materials used in the aforementioned treatments [1,2,3]. Out of the many beneficial characteristics of CSCs, the biocompatibility and hard tissue induction abilities of these materials have resulted in better cell and tissue responses to these materials [2,4].

Out of their many applications, vital pulp therapy, such as pulp capping and partial pulpotomy, require direct contact between CSCs and the dental pulp. These procedures are usually performed during pulp exposure as a result of caries excavation or traumatic events. Since pulp capping and partial pulpotomy allow the preservation of vital pulp tissue, these treatment procedures are preferred to conventional root canal treatment when appropriate indications are presented [5]. Previous studies have shown successful outcomes of the use of CSCs as pulp capping materials [3,6]. CSCs resulted in lower inflammatory response and improved dentin bridge formation compared to calcium hydroxide, a traditional pulp capping material [7,8]. Additionally, clinical studies have shown comparable or more successful outcomes following the use of CSCs as pulp capping materials compared to conventional materials such as calcium hydroxide [6,7].

In general, direct pulp capping is indicated when minor exposure of the pulp has occurred with no irreversible changes in the pulp status, which is usually determined using clinical symptoms as well as the tendency to hemorrhage during treatment procedure. However, previous studies reported that the clinical diagnosis and histologic diagnosis of the dental pulp are often different [9,10,11,12]. In fact, many teeth with caries or previous restorations, which appear to be asymptomatic with normal responses to clinical tests, presented some degree of inflammation [11]. In addition, histologic analysis has shown that some teeth with moderate caries and almost all teeth with deeply seated caries show degrees of inflammatory changes within the pulp tissue [11]. Therefore, it can be speculated that during vital pulp therapy, pulp capping materials are more likely to be placed in contact with inflamed pulp tissues. 

Although the effect of CSCs on dental pulp tissue, or dental pulp stem cells, has been of much interest, most studies on the effect of CSCs on human dental pulp stem cells (hDPSCs) have used hDPSCs without inflammation. Therefore, the aim of this study was to investigate the inflammatory cytokine production and odontogenic potential of lipopolysaccharide (LPS)-induced dental pulp stem cells (LDPSCs) using different CSCs.

## 2. Materials and Methods

### 2.1. Preparation of Material Extracts

ProRoot MTA (Dentsply, Tulsa, OK, USA), Biodentine (Septodont, Saint-Maur-des-Fossés, France), Retro MTA (Bio MTA, Seoul, Korea), and Dycal (Dentsply Caulk, Milford, DE, USA) were mixed according to the manufacturer’s instructions under aseptic conditions. The mixed materials were incubated for 24 h in a humidified atmosphere of 5% CO_2_ at 37 °C to allow setting. After setting, the materials were disintegrated into fine powder with metal beads using TissueLyser Ⅱ (Qiagen, Hilden, Germany). The material powder was mixed with complete medium (DMEM, Gibco, Grand Island, NY, USA), 1% penicillin-streptomycin (Gibco), and 10% fetal bovine serum (Gibco) at a concentration of 1 mg/ml for material extraction and incubated for 24 h in a humidified atmosphere of 5% CO_2_ at 37 °C. After 24 h of incubation, the supernatant was filtered using 0.20 μm filters (Minisart; Sartorius Stedim Biotech, Goettingen, Germany).

### 2.2. Cell Isolation and Culture

Freshly extracted, caries-free third molars of healthy patients were rinsed with wash medium (DMEM; Gibco) and 3% penicillin-streptomycin (Gibco) immediately after extraction. All procedures were conducted after obtaining informed consent. Additionally, the experimental protocol was approved by the Institutional Review Board of Yonsei University Dental Hospital (Institutional Review Board number: 2-2017-0002). The hDPSCs were obtained by using a method described in a previous study [13]. Briefly, the pulp tissues were instantly separated from the teeth and washed two times with the same wash medium mentioned earlier. Subsequently, the separated pulp tissues were minced with sterilized micro-scissors to fine pieces of approximately 0.5 mm in length, and were seeded in 6-well cell culture plates (SPL, Gyeonggi-do, Korea) with complete culture medium. The hDPSCs were cultured until subconfluence in a humidified atmosphere of 5% CO_2_ at 37 °C. Passage 4 hDPSCs were used in this study. 

### 2.3. LPS Stimulation

The hDPSCs were seeded in 100-mm culture dishes at a density of 5 × 10^5^ cells per dish with complete culture medium. After 24 h, the medium was changed to serum-free DMEM supplemented with 1% penicillin-streptomycin, 200 ng/mL CD14 (Peprotech, Rocky Hill, NJ, USA, cat#110-01) and 1 μg/mL *P. gingivalis* LPS (Invitrogen, San Diego, CA, USA, cat#tlrl-pglps) in order to induce inflammation. After 24 h, the medium was changed to a material extract conditioned medium. At every experimental time point, the supernatants and cells were harvested for enzyme-linked immunosorbent assay and quantitative real-time polymerase chain reaction (qPCR). 

### 2.4. Cell Viability Test

hDPSCs were seeded in a 96-well plate (SPL) at a density of 2 × 10^4^ cells per well. After LPS stimulation, the medium was changed to a material extract conditioned medium. Cell viability was examined at 24 h and 48 h using Cell Counting Kit-8 (Dojindo Molecular Technologies, Rockville, MD, USA) according to the manufacturer’s instructions. The absorbance was measured at 450 nm using a spectrophotometer (VersaMaxMultiplate Reader, Thermo Fisher Scientific, Waltham, MA, USA). 

### 2.5. Enzyme-Linked Immunosorbent Assay

The interleukin (IL)-6, IL-8, and transforming growth factor (TGF)-β1 concentrations in the culture supernatants were measured with enzyme-linked immunosorbent assay kits (R&D Systems, Minneapolis, MN, USA) according to the manufacturer’s instructions. The plates were read at 450 nm using a spectrophotometer (VersaMaxMultiplate Reader).

### 2.6. Quantitative Real-Time Polymerase Chain Reaction (qPCR)

The expression levels of alkaline phosphatase (ALP), osteocalcin (OCN), and runt-related transcription factor 2 (RUNX2) were analyzed. The mRNA expression levels of ALP, OCN, and RUNX2 were determined using the β-actin gene as endogenous control. Isolation of mRNA was performed using the RNeasy mini kit (Qiagen) and transcription into cDNA was performed with 500 ng RNA using RevertAid First strand cDNA synthesis kit (Thermo Fisher Scientific) according to the manufacturer’s instructions. qPCR was performed with the QuantStudio 3 system (Applied Biosystems, Foster City, CA, USA) using the following Taqman gene expression assays (Applied Biosystems): β-actin; hs01060665_g1, ALP; hs-00758162_m1, RUNX2; hs01047973_m1, OCN; and hs01587814_g1. The expression levels of target genes were calculated using the 2^−ΔΔCt^ method. 

### 2.7. Alizarin Red S (ARS) Staining

hDPSCs were seeded in a 6-well plate (SPL) at a density of 1 × 10^5^ cells per well. After LPS stimulation, the medium was changed to material extract medium with 100 mmol/L L-Ascorbic acid 2-phosphate (Sigma-Aldrich, St. Louis, MO, USA), 9 mmol/L KH_2_PO_4_ (Wako, Osaka, Japan), 10 mmol/L β-glycerol phosphate (Sigma-Aldrich), and 9.8 nmol/L dexamethasone (Sigma-Aldrich) for osteogenic induction. After 14 days, specimens were fixed with 4% paraformaldehyde (Tech-innovation, Gangwon-do, Korea) and incubated with 2% ARS solution (Acros, Gyeonggi-do, Korea) at a pH of 4.2 for 5 min at room temperature. The specimens were washed with distilled water and photographed using a digital camera (Nikon, Tokyo, Japan) and optical microscope (H.K 3.1, Koptic, Gyeonggi-do, Korea) at 40× magnification. 

### 2.8. Inductively Coupled Plasma Mass Spectrometry (ICP-MS)

Production of material extracts, cell culture, and LPS stimulation were performed using the abovementioned procedures. From the material extract conditioned medium, silicon (Si) and calcium (Ca) ion concentrations were analyzed using a 7900 inductively coupled plasma mass spectrometer (Agilent, Santa Clara, CA, USA) at 0 h, 24 h, and 48 h.

### 2.9. Statistical Analysis

Differences among groups were analyzed with one-way analysis of variance followed by the Tukey’s test using the SPSS Statistical Software version 25 (IBM Corp., Armonk, NY, USA). *p* < 0.05 was considered to indicate statistical significance. 

## 3. Results

### 3.1. Effect of Different CSCs on Cell Viability of LDPSCs

To investigate the effect of different CSCs on the cell viability of LDPSCs, cell viabilities of LDPSCs cultured with different materials were evaluated. The cell viabilities are shown in Figure 1. LPS stimulation resulted in significantly lower cell viability compared to the hDPSC control. ProRoot MTA, Biodentine, and Retro MTA did not affect the cell viability of LDPSCs, whereas Dycal significantly decreased the cell viability of LDPSCs on Day 2. 

### 3.2. Effect of Different CSCs on the Inflammatory Responses of LDPSCs

The expressions of IL-6 and IL-8, which are pro-inflammatory cytokines, as well as of TGF-β1, an anti-inflammatory cytokine, are shown in Figure 2. The expressions of IL-6 and IL-8 were significantly increased following LPS treatment. No material had any significant effect on the production of IL-6 and IL-8 by LDPSCs after 12 and 24 h. After 48 h, Retro MTA significantly decreased the expression of IL-6 and IL-8 by LDPSCs. ProRoot MTA, Biodentine, and Dycal did not have any significant effect (Figure 2A,B). 

LPS treatment on hDPSCs resulted in a significant decrease in TGF-β1 production at all times. ProRoot MTA and Biodentine significantly decreased TGF-β1 expression by LDPSCs after 12 h and 48 h, respectively. Retro MTA and Dycal did not have significant effect on the TGF-β1 expression by LDPSCs (Figure 2C). 

### 3.3. Effect of Different CSCs on the Odontogenic Differentiation of LDPSCs

To evaluate the effect of CSCs on the odontogenic differentiation of LDPSCs, ALP, OCN, and RUNX2 mRNA expressions were measured. No CSC had a significant effect on the ALP expression by LDPSCs (Figure 3A). Regarding OCN expression by LDPSCs, Biodentine resulted in a significant increase after 12 h. Retro MTA also significantly increased OCN expression after 12 h and 24 h (Figure 3B). All the CSCs significantly increased RUNX2 expression by LDPSCs after 12 h. The differences were not statistically significant at other time points (Figure 3C). 

ARS staining performed after 14 days is shown in Figure 4. LPS stimulation of hDPSCs resulted in increased formation of calcium deposits. LDPSCs cultured with Biodentine extract medium presented significantly greater calcium deposition compared to other groups. LDPSCs with Dycal extract medium displayed significantly decreased ARS stained areas. 

### 3.4. Si and Ca Ion Concentrations from Material Extract Medium

The concentrations of Si and Ca ions were measured using ICP-MS. Compared to other material extract media, ProRoot MTA extract medium showed greatly increased concentrations of Si ions at 24 h and 48 h. Biodentine, Retro MTA, and Dycal also showed differing levels and trends of Si ion concentrations (Figure 5A). Regarding Ca ion levels, ProRoot MTA, Biodentine, and Retro MTA extract media showed similar levels and patterns of Ca ion concentrations. Dycal extract medium, however, showed lower levels of Ca ions compared to other materials (Figure 5B).

## 4. Discussion

Direct pulp capping is recommended when minor pulp exposure occurs after removal of tooth structure, and many studies have reported successful outcomes following this treatment [5,14,15]. The material used to seal the exposed pulp surface is crucial for a successful outcome [6,16]. The biocompatibility, inflammatory effects, and odontogenic potential of the capping material are especially critical to avoid degeneration and inflammatory responses of the pulp that may lead to irreversible changes in the pulp status [16]. Previous studies on healthy hDPSCs have shown superior biocompatibility and odontogenic potential as well as reduced inflammatory reactions when CSCs were used as pulp capping material [17]. 

Four pulp capping materials were used in this study: ProRoot MTA, Biodentine, Retro MTA, and Dycal. Whereas ProRoot MTA, Biodentine, and Retro MTA are CSCs, Dycal is a conventional pulp capping material mainly consisting of Ca(OH)_2_. Previous studies have shown superior outcomes when ProRoot MTA was used in direct pulp capping compared to Ca(OH)_2_ [6]. Histologic studies have also shown more homogenous dentin bridge formation and mild or almost absent inflammation with ProRoot MTA as capping material [16,18].

In our study, the effects of CSCs on LDPSCs, instead of normal hDPSCs, were evaluated. Traditionally, pulp capping was performed based on patients’ initial symptoms and degree of hemorrhage from pulp tissue. However, since the clinical and actual pathological diagnoses can differ [11], it is highly likely that the exposed pulp surface in contact with CSCs may undergo a certain degree of inflammation. In addition, recent studies have reported successful outcomes of vital pulp therapy even when the teeth were diagnosed with “irreversible” pulpitis based on initial clinical symptoms [19,20]. These findings suggest the possibility that indications for vital pulp therapy may be extended in the future, which necessitates studies on the effect of pulp capping materials on inflamed dental pulp stem cells.

Inflammatory response was evaluated by measurement of the cytokines IL-6, IL-8, and TGF-β1. Previous studies have shown that treatment of hDPSCs with LPS results in the expression of IL-6 and IL-8 [21,22], and a recent study reported that LPS and dentin matrix proteins may have interactive effects on the immunomodulatory effects of hDPSCs [23]. TGF-β1 is known to have anti-inflammatory effects [24] and is involved in the repair events after pulp inflammation [25]. It plays diverse roles in the human dental pulp including cell proliferation and migration, extracellular matrix production as well as induction of odontoblastic differentiation. In our study, ProRoot MTA and Biodentine did not have any significant effect on the expression of IL-6 and IL-8 by LDPSCs, whereas Retro MTA significantly decreased IL-6 and IL-8 expression after 48 h. In addition, ProRoot MTA and Biodentine significantly lowered TGF-β1 expression by LDPSCs. However, Retro MTA and Dycal did not have a significant effect on TGF-β1 expression by LDPSCs.

Although studies on the inflammatory responses of LDPSCs are limited, one study reported decreased production of pro-inflammatory cytokines when LDPSCs were cultured with ProRoot MTA and Endocem MTA [26]. On the other hand, another study reported increased production of IL-1β when LDPSCs were cultured with ProRoot MTA [27]. The inflammatory response of LDPSCs can differ based on the type of CSC, due to the difference in composition and ion release between the materials [27]. In our study, the results were different between the three CSCs, and only Retro MTA significantly lowered IL-6 and IL-8 expressions. Lai et al. discussed that the Ca and Si ion release from CSCs can differ based on the composition and solubility of the material [27]. Although Ca and Si ion concentrations are known to be critical in cell behavior, including cell attachment, proliferation, and differentiation [28], excess of these ions may induce cell apoptosis [29,30]. Increased release of Si ions from CSCs, most likely from the SiO_2_ phase, may result in hyperosmoticity, inducing the production of inflammatory cytokines [31]. 

We evaluated the Si ion concentrations of the material extract media, and ProRoot MTA showed significantly higher levels of Si ion concentrations at all time points compared to the other materials. Biodentine, Retro MTA, and Dycal also showed differing levels and patterns of Si ion release. The difference in material composition may account for the varying ion concentrations of materials. Whereas ProRoot MTA is known to contain 21.1 wt % of SiO_2_ [32], Retro MTA contains 5–15 wt % SiO_2_. Moreover, Biodentine mainly consists of tricalcium silicate and dicalcium silicate. Such differences in ion concentration may have affected the production of IL-6, IL-8, and TGF-β1 among the CSCs. 

Odonto/osteogenic differentiation is also essential since it allows the formation of a hard tissue barrier in the pulp-capped area. One previous study evaluated the odontogenic differentiation of LDPSCs using extracts of ProRoot MTA, Endocem MTA, and Dycal [26]. The study showed increased dentin matrix protein 1 and dentin sialophosphoprotein levels with the presence of all three materials. In the present study, ALP, OCN, and RUNX2 expressions were evaluated, and CSCs had different effects on each protein No CSC had a significant effect on the ALP expression of LDPSCs. However, Biodentine and Retro MTA significantly increased OCN expression. All CSCs increased RUNX2 expression by LDPSCs after 12 h, but there was no significant difference at other time points. The fact that genes play diverse roles in different stages of osteoblast differentiation may be a notable factor for the differences in gene expressions. ARS staining results showed increased stains of calcium deposits for Biodentine compared to the other material extracts. 

The release of Ca ions from pulp capping materials is crucial for osteogenic differentiation and mineralization. However, different osteogenic markers show varying responses to the increase of Ca ions. A previous study reported reduced ALP and RUNX2 gene levels at elevated Ca ion concentrations, while OCN levels increased [33]. In addition, no linear relationship exists between Ca ion concentration and the onset of mineralization because, when the extracellular Ca ion concentration is above optimal level, cytotoxic effects may outweigh any osteogenic potential [34]. According to previous studies, CSCs show faster and more abundant initial release of Ca ions compared to conventional pulp capping materials [35,36]. Also in our study, CSCs showed higher Ca ion concentrations compared to Dycal. Different CSCs may show varying patterns and extracellular concentrations of Ca ions. However, although CSCs in the present study showed similar levels and trends of Ca ion concentrations to a previous study [27], there were no clear differences between the Ca ion concentrations of CSCs. 

Dycal significantly increased ALP, OCN, and RUNX2 levels and resulted in higher expressions of all odontogenic markers compared to CSCs. This is not in accordance with previous findings that demonstrated a higher odonto/osteogenic potential of CSCs than Ca(OH)_2_ [2]. An explanation may be that LPS-treated inflamed hDPSCs were used in the present study. However, previous in vivo studies have shown more homogenous and continuous dentin bridge formation in pulp capping with CSCs, whereas Ca(OH)_2_ showed heterogeneous, porous dentin bridges with cell inclusions [37]. Since the quality of the dentin bridge is more essential than its mere thickness or quantity, higher expression of osteogenic genes may not necessarily indicate the formation of intact dentin bridges. 

## 5. Conclusions

In conclusion, ProRoot MTA, Biodentine, Retro MTA, and Dycal resulted in different levels of inflammatory cytokines and odonto/osteogenic markers in LPS-stimulated hDPSCs. Retro MTA decreased the production of IL-6 and IL-8 and had no effect on TGF-β1 levels. Other CSCs had no significant effect on IL-6 and IL-8 production but decreased TGF-β1 levels. Regarding odontogenic markers, no CSC had a significant effect on ALP expression. However, CSCs increased RUNX2 levels of LDPSCs after 12 h. Biodentine and Retro MTA also significantly increased the expression of OCN.

## Figures and Tables

**Figure 1 materials-12-01259-f001:**
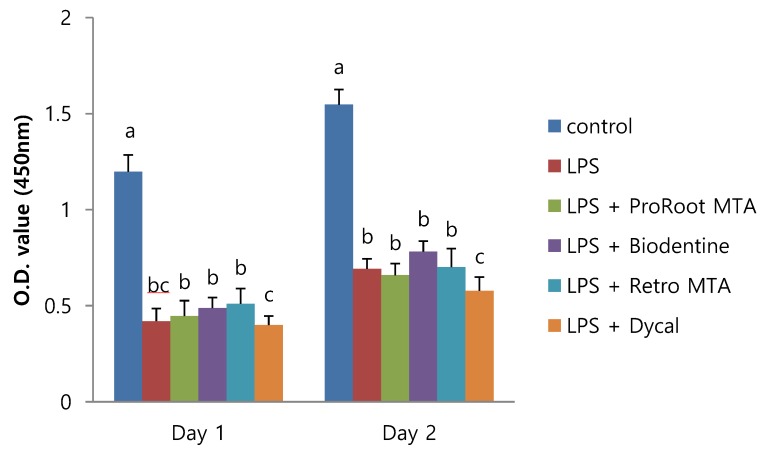
The effects of calcium silicate cements (CSCs) on the cell viability of human dental pulp stem cells (hDPSCs) and LPS-induced dental pulp stem cells (LDPSCs). Absorbance values were measured at 450 nm using the Cell Counting Kit-8 assay. a > b, c; b > c (*p* < 0.05), different alphabet letters represent statistically significant differences between the materials within the same time group. Control, hDPSCs cultured with normal medium; LPS, LDPSCs cultured with normal medium; LPS + ProRoot MTA, LDPSCs cultured with ProRoot MTA extract medium; LPS + Biodentine, LDPSCs cultured with Biodentine extract medium; LPS + Retro MTA, LDPSCs cultured with Retro MTA extract medium; LPS + Dycal, LDPSCs cultured with Dycal extract medium. O.D., optical density.

**Figure 2 materials-12-01259-f002:**
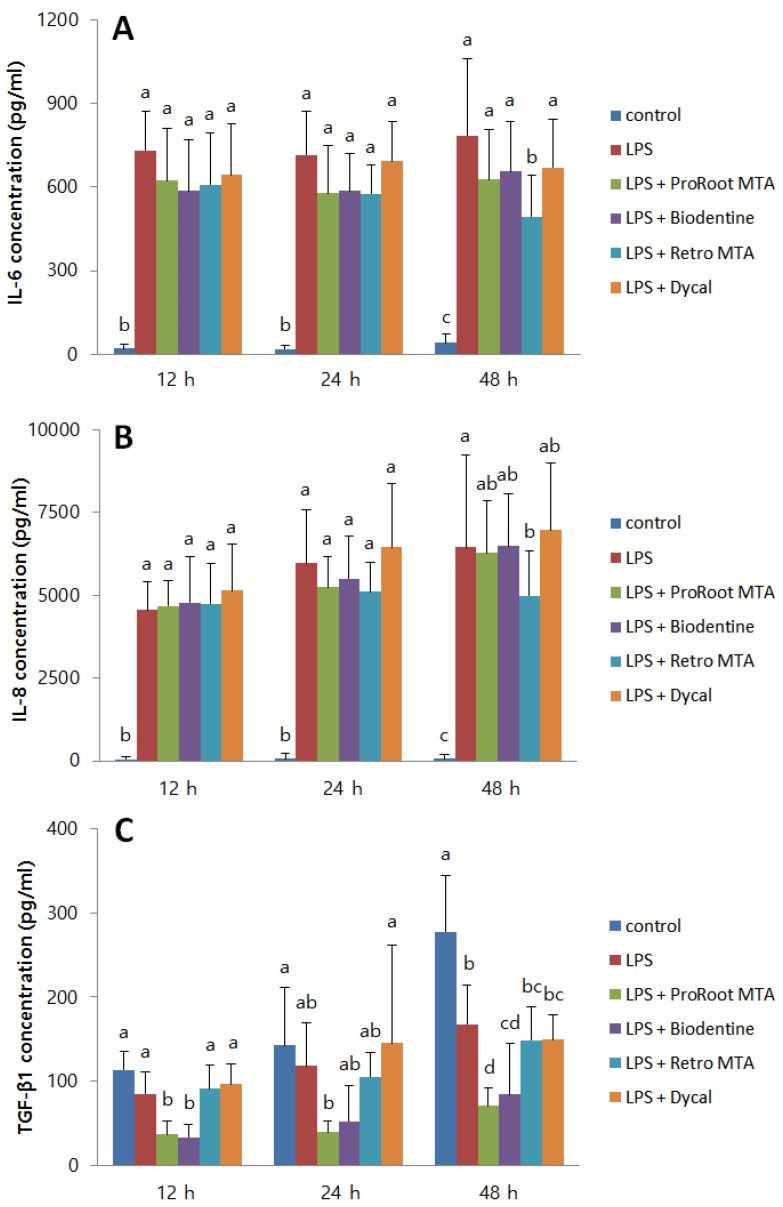
The effects of CSCs on the inflammatory response of LDPSCs. (**A**) IL-6, (**B**) IL-8, and (**C**) TGF-β1 expressions were measured using the enzyme-linked immunosorbent assay. (**A**) IL-6: a > b, c; b > c (*p* < 0.05), (**B**) IL-8: a > b, c; b > c (*p* < 0.05), (**C**) TGF-β1: a > b, c, d; b > c, d; c > d (*p* < 0.05), different alphabet letters represent statistically significant differences between the materials within the same time group. Control, hDPSCs cultured with normal medium; LPS, LDPSCs cultured with normal medium; LPS + ProRoot MTA, LDPSCs cultured with ProRoot MTA extract medium; LPS + Biodentine, LDPSCs cultured with Biodentine extract medium; LPS + Retro MTA, LDPSCs cultured with Retro MTA extract medium; LPS + Dycal, LDPSCs cultured with Dycal extract medium.

**Figure 3 materials-12-01259-f003:**
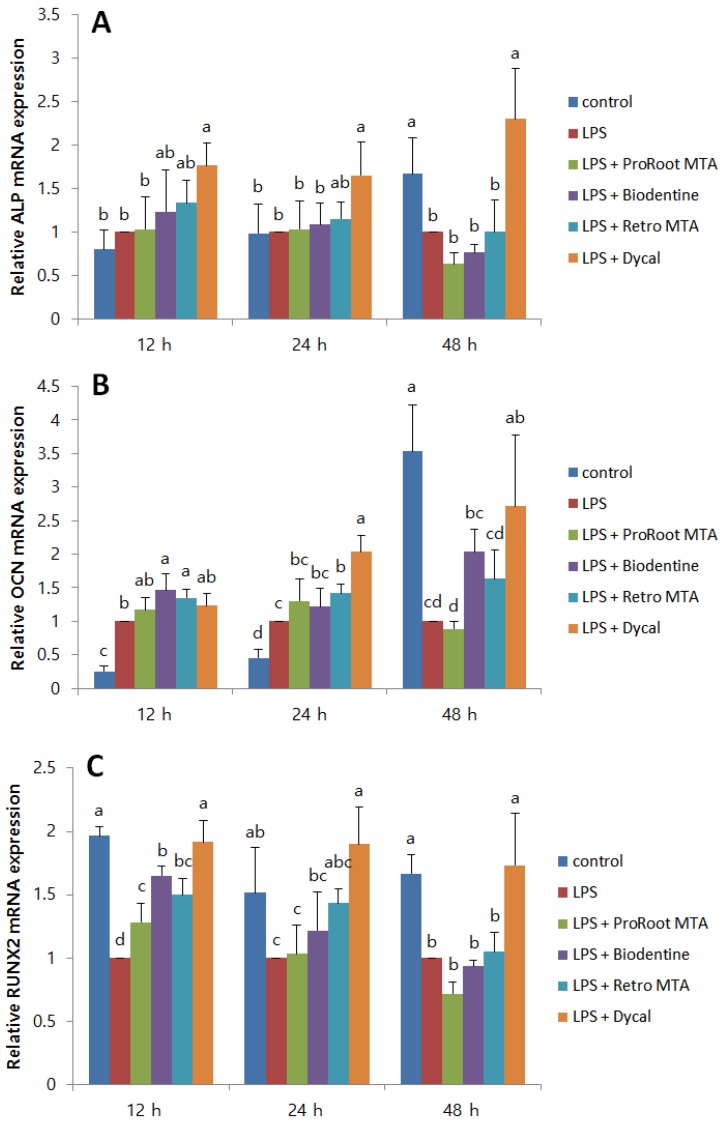
The effects of CSCs on the ALP, OCN, and RUNX2 expressions of LDPSCs. Relative (**A**) ALP (**B**) OCN, and (**C**) RUNX2 expressions were measured using quantitative real-time polymerase chain reaction (qPCR) and normalized using β-actin. (**A**) ALP: a > b (*p* < 0.05), (**B**) OCN: a > b, c, d; b > c, d; c > d (*p* < 0.05), (**C**) RUNX2: a > b, c, d; b > c, d; c > d (*p* < 0.05), different alphabet letters represent statistically significant differences between the materials within the same time group. Control, hDPSCs cultured with normal medium; LPS, LDPSCs cultured with normal medium; LPS + ProRoot MTA, LDPSCs cultured with ProRoot MTA extract medium; LPS + Biodentine, LDPSCs cultured with Biodentine extract medium; LPS + Retro MTA, LDPSCs cultured with Retro MTA extract medium; LPS + Dycal, LDPSCs cultured with Dycal extract medium.

**Figure 4 materials-12-01259-f004:**
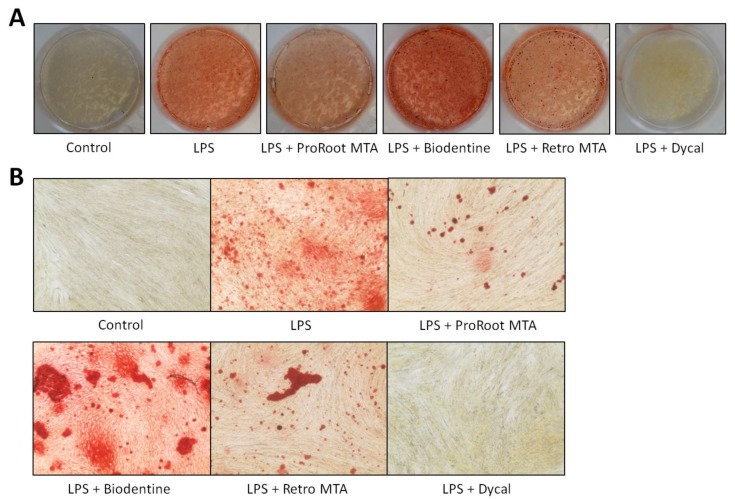
The effects of CSCs on the odontogenic differentiation of LDPSCs. ARS staining was performed after 14 days. (**A**) Photographs of the experimental wells; (**B**) microscope images of 40× magnification are shown. Control, hDPSCs cultured with normal medium; LPS, LDPSCs cultured with normal medium; LPS + ProRoot MTA, LDPSCs cultured with ProRoot MTA extract medium; LPS + Biodentine, LDPSCs cultured with Biodentine extract medium; LPS + Retro MTA, LDPSCs cultured with Retro MTA extract medium; LPS + Dycal, LDPSCs cultured with Dycal extract medium.

**Figure 5 materials-12-01259-f005:**
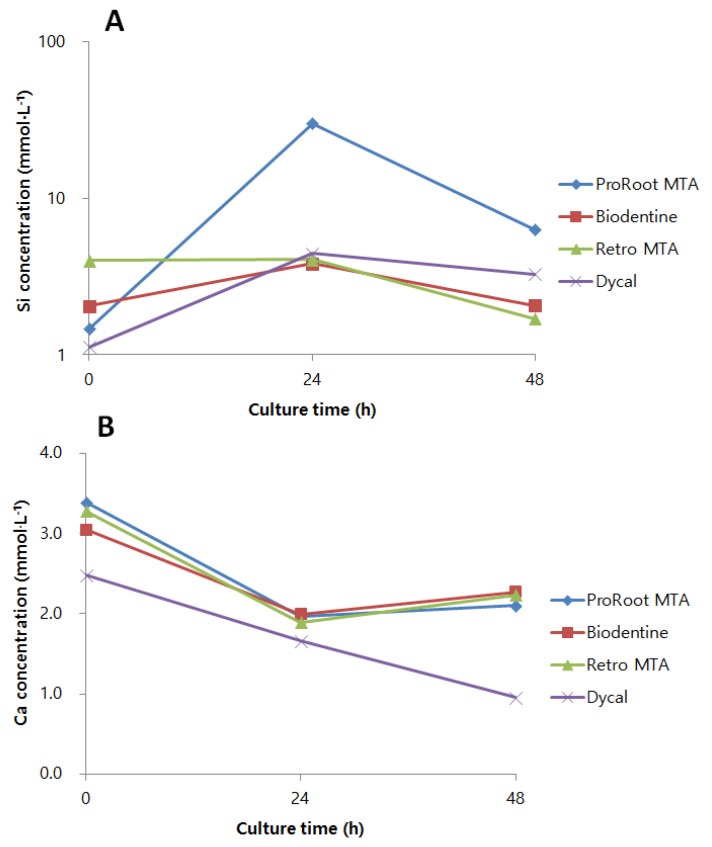
Ion concentrations from material extract medium evaluated by ICP-MS. (**A**) Si and (**B**) Ca ion concentrations of material extract-conditioned medium were measured up to 48 h. ProRoot MTA, LDPSCs cultured with ProRoot MTA extract medium; Biodentine, LDPSCs cultured with Biodentine extract medium; Retro MTA, LDPSCs cultured with Retro MTA extract medium; Dycal, LDPSCs cultured with Dycal extract medium.

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
