# Peer review of "Effects of Different Calcium Silicate Cements on the Inflammatory Response and Odontogenic Differentiation of Lipopolysaccharide-Stimulated Human Dental Pulp Stem Cells"

_materials, 2019, doi:10.3390/ma12081259_

Reviewer 1 Report

Interesting study to compare the effects of commercial pulp capping materials on the inflammatory pulp cells. The limited results were describe well and bring out some pretty good discussions. Comments are as follows:

How did you determine the materials extration concentration of 1 mg/ml? When apply these materials to pulp the concentration in local and distal side should be different. Therefore, different ratios of the extraxt medium (ie. 100%, 50%, 10%) should be compared in the same time to understand their effects on the cells.

The authors explain that the regulation of inflamation cytokine were due to the different Ca, Si... ions levels. However, it is very easy to measure the released ions in the extraction medium by AAS or ICP and can give straight evidence to the results.

Author Response

1.     How did you determine the materials extraction concentration of 1 mg/ml? When apply these materials to pulp the concentration in local and distal side should be different. Therefore, different ratios of the extract medium (ie. 100%, 50%, 10%) should be compared in the same time to understand their effects on the cells.

-> We certainly agree testing different concentrations of extract medium is necessary. Before starting our main experiment, we performed cell viability tests with different concentrations of material extract media, and found 1mg/ml to be the most appropriate concentration for our experimental design. 

2.     The authors explain that the regulation of inflammation cytokine were due to the different Ca, Si... ions levels. However, it is very easy to measure the released ions in the extraction medium by AAS or ICP and can give straight evidence to the results.

-> Thank you for your valuable comment. We performed inductively coupled plasma mass spectrometry (ICP-MS) to observe the Si and Ca ion concentration level changes from the material extract conditioned medium. The results are included in the manuscript (Figure 5).

Reviewer 2 Report

The manuscript “Effects of Different Calcium Silicate Cements on the Inflammatory Response and Odontogenic Differentiation of Lipopolysaccharide-stimulated Human Dental Pulp Stem Cells” is under the scope of this journal, well-written and adds knowledge to the field.

The references are mostly up-to-date and appropriately cited along the manuscript.

This reviewer only has 2 suggestions for authors to improve the quality and actuality of the paper:

P1L44-46 – Instead of citing only narrative reviews about this issue, consider some original research studies (Sequeira, DB et al. Effects of a New Bioceramic Material on Human

Apical Papilla Cells. Journal of Functional Biomaterials 2018);

P2L60 – References 8 to 10 can be replaced or accompanied by a more recent and relevant study about this issue (Ricucci, D et al. Correlation between Clinical and Histologic Pulp Diagnoses. Journal of Endodontics 2014).

Author Response

P1L44-46 – Instead of citing only narrative reviews about this issue, consider some original research studies (Sequeira, DB et al. Effects of a New Bioceramic Material on Human Apical Papilla Cells. Journal of Functional Biomaterials 2018);

P2L60 – References 8 to 10 can be replaced or accompanied by a more recent and relevant study about this issue (Ricucci, D et al. Correlation between Clinical and Histologic Pulp Diagnoses. Journal of Endodontics 2014).

è As suggested, we have included the two studies in the manuscript. Thank you for your suggestions.

Reviewer 3 Report

Please add the following reference and include respective aspect in the discussion as it is a recent and valuable paper in this field of research:

Widbiller, M., Eidt, A., Wölflick, M., Lindner, S. R., Schweikl, H., Hiller, K.-A., et al. (2018). Interactive effects of LPS and dentine matrix proteins on human dental pulp stem cells. International Endodontic Journal, 51(8), 877–888. http://doi.org/10.1111/iej.12897

Author Response

1.     Please add the following reference and include respective aspect in the discussion as it is a recent and valuable paper in this field of research:

Widbiller, M., Eidt, A., Wölflick, M., Lindner, S. R., Schweikl, H., Hiller, K.-A., et al. (2018). Interactive effects of LPS and dentine matrix proteins on human dental pulp stem cells. International Endodontic Journal, 51(8), 877–888. http://doi.org/10.1111/iej.12897

è Thank you for proposing a valuable study to be noted. We have included the recommended paper in the reference and discussion sections.

Reviewer 4 Report

In this paper, the authors investigated the effects of different calcium silicate cements (CSCs, including ProRoot MTA, Biodentine, Retro MTA, and Dycal) on the production of inflammatory cytokines and mRNA expression of odontogenic differentiation-related molecules in LPS-stimulated human dental pulp stem cells (LDPSCs). Their results showed that, only Dycal decreased the cell viability of LDPSCs for up to 48 hours. Retro MTA significantly decreased the expression of IL-6 and IL-8 in LDPSCs. ProRoot MTA and Biodentine significantly reduced TGF-β expression in LDPSCs. All CSCs had no significant effects on the expression of odontogenic molecules. This is an interesting study. But the research design has some weakness. The authors also showed improve the presentation. My concerns about this manuscript are as follows.

1.     Dental pulp is a complex tissue. There are many types of cells, such as fibroblasts, neuronal cells, local macrophages, epithelial cells, etc. The inflammatory response involves a highly coordinated network of many cell types. Activated macrophages, monocytes, and other cells mediate local responses to tissue damage and infection (Chen L, Deng H, Cui H, et al. Inflammatory responses and inflammation-associated diseases in organs. Oncotarget. 2017;9(6):7204-7218. Published 2017 Dec 14. doi:10.18632/oncotarget.23208). Therefore, the authors should check the effects of CSCs on LPS-induced inflammatory response in other type of cells or perform some in vivo experiment in animal models.

2.     Odontogenic differentiation of LDPSCs should use inductive medium. ALP, OCN and RUNX mRNA level can’t represent odontogenic differentiation. The authors at least should perform Alizarin red staining.

3.     Meaning of alphabet letters in all figures is unclear. What is the p value of each alphabet? Compared with which group?

Author Response

1.     Dental pulp is a complex tissue. There are many types of cells, such as fibroblasts, neuronal cells, local macrophages, epithelial cells, etc. The inflammatory response involves a highly coordinated network of many cell types. Activated macrophages, monocytes, and other cells mediate local responses to tissue damage and infection (Chen L, Deng H, Cui H, et al. Inflammatory responses and inflammation-associated diseases in organs. Oncotarget. 2017;9(6):7204-7218. Published 2017 Dec 14. doi:10.18632/oncotarget.23208). Therefore, the authors should check the effects of CSCs on LPS-induced inflammatory response in other type of cells or perform some in vivo experiment in animal models.

è Thank you for your valuable comment. We agree that inflammatory, immunomodulatory responses include many types of cells, and the inclusion of immune cells in addition to human dental pulp stem cells (hDPSCs) may provide further knowledge of these responses. In vivo studies will definitely be helpful to expand knowledge into this topic, and we hope to design a related animal study in the future. However, we still believe our study provides meaningful understanding of the effect calcium silicate materials have on inflamed hDPSCs.

2.     Odontogenic differentiation of LDPSCs should use inductive medium. ALP, OCN and RUNX mRNA level can’t represent odontogenic differentiation. The authors at least should perform Alizarin red staining.

è As suggested, we additionally carried out Alizarin Red S Staining, and included the results in the manuscript (Figure 4). During cell culture, the supernatants were used in the enzyme-linked immunosorbent assay and the cells were used to determine the mRNA levels. Therefore, the analysis of ALP, OCN, and RUNX2 mRNA expressions were done without the use of inductive medium.

3.     Meaning of alphabet letters in all figures is unclear. What is the p value of each alphabet? Compared with which group?

è The alphabets represent statistically significant differences within the same time group. Different alphabets indicate that there is a difference between the materials within the same time group at P < 0.05. We added a description to figure legend.

Round  2

Reviewer 1 Report

The authors provide ICP and cell ARS staining results and make some good discussion.

Author Response

Once again, thank you for your comment.

Reviewer 4 Report

The authors answered most of my questions, performed Aizarin Red O staining and measured Si and Ca ion concentrations from material extract medium. They did good job. But they still need to define the meaning of alphabet letters in figures. They should use “*, **, ***” to show the significance within the group and between groups.  Please see the example from the below website. http://www.jnsbm.org/viewimage.asp?img=JNatScBiolMed_2012_3_2_125_101879_f5.jpg  

Author Response

Thank you for your valuable comment.

We agree your recommendation is an effective way to present data and tried to change the figures as you suggested. In our study, however, there are a lot of different groups with various significance, and by changing method to present significance, the graphs seemed to get too perplexing. 

Using alphabets seemed to make the figures more understandable, so we decided to change the order of alphabets to make the figures more straight-forward. The figure legends were also adjusted.

This manuscript is a resubmission of an earlier submission. The following is a list of the peer review reports and author responses from that submission.